# Asymmetric power dissipation in electronic transport through a quantum point contact

**Carmen Blaas-Anselmi, Félix Helluin, Rodolfo A. Jalabert, Guillaume Weick and Dietmar Weinmann⋆**

Université de Strasbourg, CNRS, Institut de Physique et Chimie des Matériaux de Strasbourg, UMR 7504, F-67000 Strasbourg, France

⋆ dietmar.weinmann@ipcms.unistra.fr

## Abstract

We investigate the power dissipated by an electronic current flowing through a quantum point contact in a two-dimensional electron gas. Based on the Landauer-Büttiker approach to quantum transport, we evaluate the power that is dissipated on the two sides of the constriction as a function of the Fermi energy, temperature, and applied voltage. We demonstrate that an asymmetry appears in the dissipation, which is most pronounced when the quantum point contact is tuned to a conductance step where the transmission strongly depends on energy. At low temperatures, the asymmetry is enhanced when the temperature increases. An estimation for the position of the maximum dissipation is provided.

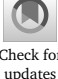
# 1 Introduction

The Landauer-Büttiker approach describes quantum transport through a mesoscopic device by phase-coherent elastic scattering [1,2]. The electrical conductance of the device is determined by the quantum transmission of electrons from one attached electrode to another one through the mesoscopic sample in which the electrons can be scattered. The connection between the sample and the electrodes is thought to be realized via perfect quasi-one-dimensional leads. The electrodes are assumed to be large reservoirs, with fixed temperatures and electrochemical potentials, unaffected by outgoing or incoming electrons. The presence of macroscopic reservoirs allows one to approach the thermodynamic limit of an ideal heat bath and particle reservoir with a continuous spectrum and the possibility of energy dissipation.

Within the idealized scheme of sample/leads/reservoirs, the energy dissipation occurs through thermalization of the traveling electrons within the reservoirs [3–5]. However, in actual physical systems the previous scheme is not so clear-cut, since the separation between leads and reservoir is somehow arbitrary, and the question of where the dissipation takes place becomes relevant.

From the experimental point of view, addressing the previous stationary, out of equilibrium phenomenon necessitates, in addition to the global measurement of the conductance, the development of local probes.

In mesoscopic metallic wires a tunnel superconducting probe allowed to determine the energy distribution of Landau quasiparticles between two reservoir electrodes [6]. For short wires ($1.5\,\mu$m long) the energy distribution in the middle of the sample is given by the half sum of the two Fermi distributions of the reservoirs, indicating that the dissipation occurs indeed in the reservoirs and that the scattering within the wire is almost elastic. On the contrary, for longer wires ($5\,\mu$m long), away from the electrodes, the electron distribution approaches a thermal one, indicating that the diffusing quasiparticles within the sample thermalize through the residual electron-electron interaction.

In the paradigmatic case of a ballistic quantum point contact (QPC), the distinction between leads and reservoirs is not obvious, and the accepted view is that the transition from the first to the second element occurs at approximately a distance from the constriction defined by the phase-coherence length $L_\Phi$. Thus, the current description and understanding of conductance quantization in a QPC stems from the theoretical framework settled by the scattering approach.

Local probes, like the scanning gate microscopy (SGM) allow getting further information about electronic transport than that yielded by the measurement of the conductance [7–9]. In particular, the behavior of the electron flow, weakly or strongly perturbed by the scanning tip, could be analyzed within a semiclassical framework in terms of classical trajectories of noninteracting electrons in a weak and smoothly disordered landscape. However, the above-mentioned dissipation issues cannot be addressed with such a technique.

Scanning thermal microscopy (SThM) provides a probe of the local temperature, yielding access to the question of how and where dissipation takes place. In particular, the development of a SQUID-on-tip thermometer has recently achieved a nanoscale spatial resolution ($50-100$ nm) with a $\mu$K sensitivity [10–12], and allowed to observe a remarkable spatial separation between where the voltage drops (the resistance) and where the associated Joule heating (dissipation) occurs.

Our work is motivated by the possibility to detect local temperature changes caused by an electronic current flow in the quantum transport regime. Of particular interest are measurements of the local heating close to a QPC with and without magnetic field which put in evidence a spatial asymmetry of the dissipation [13]. On the side of the QPC with lower electrochemical potential, the dissipated heat is generated by the transmitted electrons which

thermalize from energies above the Fermi sea. On the opposite side, the transport process mainly leaves holes in the Fermi sea that thermalize by moving to the surface. For an energy-independent transmission probability of the QPC, particle-hole symmetry leads to symmetric power dissipation on the two sides of the QPC [14]. Conversely, experiments on atomic scale junctions, complemented by a scattering approach with the transmission obtained from *ab-initio* calculations [15,16] have yielded asymmetries in the heat dissipation of systems where the transmission is strongly energy-dependent. Such a heat asymmetry has been shown to be reduced by inelastic and dephasing effects [17]. Studies based on a hydrodynamic model of charge and heat flow in inhomogeneous two-dimensional electron systems have also found asymmetric heat dissipation [18].

In this paper, working within the Landauer-Büttiker formalism of quantum transport, we calculate the difference between the dissipated power on the two sides of the QPC. We determine the asymmetry of the dissipation when the transmission through the QPC varies in the energy range between the two chemical potentials corresponding to conductance quantization and conductance steps. Moreover, we study the dependence of the appearing asymmetry on the relevant system parameters (Fermi energy, bias voltage, and temperature) and provide an estimate for the distance from the QPC to the point of maximum dissipation.

This paper is outlined as follows: In Sec. 2 we present the general description of the energy dissipation within the Landauer-Büttiker formalism of electronic transport. The theory is applied in Sec. 3 to a model of a QPC formed by an abrupt constriction in a two-dimensional electron gas (2DEG). An estimate of the position of maximum dissipation is proposed in Sec. 4. Conclusions are drawn and some perspectives of our work are discussed in Sec. 5.

## 2  Energy dissipation on the two sides of a scatterer

Considering a generic mesoscopic sample, the current carried through by noninteracting electrons, from the left (L) to the right (R) reservoir is [3,4]

$$I = \frac{2e}{h} \int_{-\infty}^{\infty} d\varepsilon \, \mathcal{T}(\varepsilon, V)[f(\varepsilon - \mu_{\mathrm{L}}) - f(\varepsilon - \mu_{\mathrm{R}})]. \tag{1}$$

The equilibrium Fermi-Dirac distribution function $f(\varepsilon) = [\exp(\varepsilon/k_{\mathrm{B}}T) + 1]^{-1}$ sets the occupation at the reservoirs, assumed to have both the same temperature $T$. We note $k_{\mathrm{B}}$ the Boltzmann constant, $h$ the Planck constant, $\bar{\mu}$ the mean electrochemical potential, $V$ the applied bias voltage, and $e$ the electron charge ($e < 0$). The electrochemical potential in the left (right) reservoir is therefore $\mu_{\mathrm{L(R)}} = \bar{\mu} \pm eV/2$, and in the limit of low voltage at zero temperature, the mean electrochemical potential $\bar{\mu}$ is the Fermi energy of the system. In order to visualize the transfer of electrons from left to right, we will assume $V < 0$ (and thus $\mu_{\mathrm{L}} > \mu_{\mathrm{R}}$). The total transmission coefficient $\mathcal{T} = \sum_{a,b}^{N_{\mathrm{L}}, N_{\mathrm{R}}} |t_{ba}|^2$ is obtained as a sum over the $N_{\mathrm{L}}$ and $N_{\mathrm{R}}$ propagating channels in the L and R leads, respectively. We follow the standard notation of calling $t(t')$ and $r(r')$ the transmission and reflection submatrices of the $[N_{\mathrm{L}} + N_{\mathrm{R}}] \times [N_{\mathrm{L}} + N_{\mathrm{R}}]$ scattering matrix $S$ for particles impinging from the left and right side of the scatterer [5]. The Landauer-Büttiker zero-temperature linear conductance $(\partial I/\partial V)|_{V=0}$ follows from Eq. (1), and writes $g = (2e^2/h) \mathcal{T}(\bar{\mu})$.

Even if in writing Eq. (1) we ignored the electron-electron interaction, the energy- and bias voltage-dependent transmission coefficient is difficult to determine, as it results from a self-consistent treatment of the electron density in the applied electric field [19–21]. Establishing an ansatz on the spatial dependence of the potential drop allows to avoid the self-consistent treatment and to use a one-particle approach [22,23], which is able to account for the main

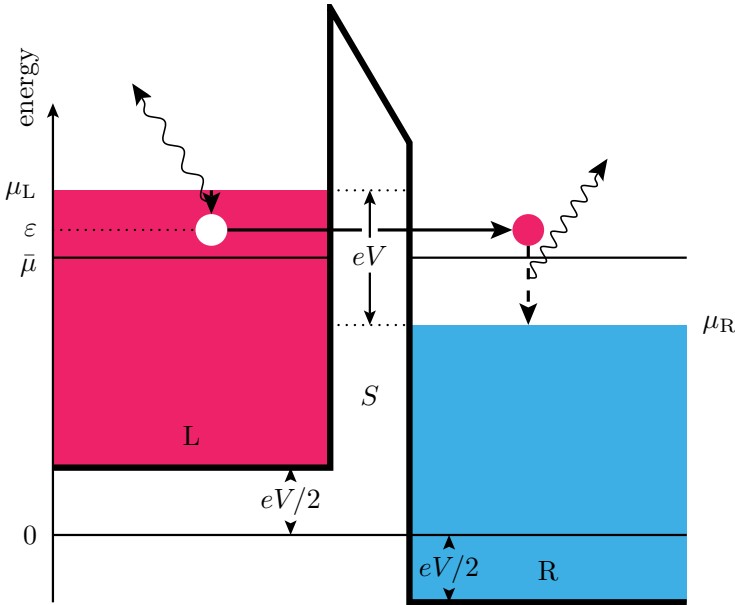

Figure 1: Scheme of the zero-temperature dissipation process associated with the elastic transmission of an electron with energy $\varepsilon$ through a generic mesoscopic sample represented by a scattering matrix $S$. For presentation purposes, the scatterer is represented as a potential barrier, and the ansatz of a linear voltage drop within the scatterer is adopted. The electron densities are taken to be equal on both sides of the scatterer, ensuring charge neutrality. A traveling electron delivers an energy $\varepsilon - \mu_R$ at the right reservoir, while the hole it leaves behind in the left reservoir releases an energy $\mu_L - \varepsilon$ when neutralized by an electron from the Fermi level. The applied bias voltage $V$ verifies $eV = \mu_L - \mu_R$, with $\mu_{L(R)}$ the electrochemical potential in the left (right) reservoir and $e$ the electron charge. The bottom of the conduction band in the unbiased case is chosen as the energy origin.

features experimentally observed in the nonlinear conductance of QPCs [24, 25]. We will establish in the sequel that, while the total power dissipation scales as $V^2$, the power asymmetry is of order $V^3$. In a systematic expansion in orders of $V$, applicable to the case of a relatively small bias voltage, the evaluation of the power asymmetry in the leading order in $V$ only requires considering the transmission coefficient to order $V^0$, and thus, we will ignore henceforth the second argument of $\mathcal{T}(\varepsilon, V)$.

## 2.1 General expressions

The dissipated power at the left (L) and right (R) of the scatterer can be expressed as

$$\mathcal{P}_{L/R} = \int_{-\infty}^{\infty} d\varepsilon \; p_{L/R}(\varepsilon), \tag{2}$$

where

$$p_L(\varepsilon) = \frac{2}{h}(\mu_L - \varepsilon)\mathcal{T}(\varepsilon)\left[f(\varepsilon - \mu_L) - f(\varepsilon - \mu_R)\right], \tag{3a}$$

$$p_R(\varepsilon) = \frac{2}{h}(\varepsilon - \mu_R)\mathcal{T}(\varepsilon)\left[f(\varepsilon - \mu_L) - f(\varepsilon - \mu_R)\right]. \tag{3b}$$

As sketched in Fig. 1, the factors $\varepsilon - \mu_R$ and $\mu_L - \varepsilon$ represent the thermalization energy of an excited electron or hole to the Fermi level in the right and left reservoirs, respectively.

From Eqs. (2) and (3) we verify that the total dissipated power is $\mathcal{P}_\mathrm{T} = \mathcal{P}_\mathrm{L} + \mathcal{P}_\mathrm{R} = VI$, in agreement with Ohm's law. The expressions for the dissipated power are related to the energy flow through the scatterer [26]

$$\mathcal{J} = \frac{2}{h} \int_{-\infty}^{\infty} \mathrm{d}\varepsilon\, \varepsilon\, \mathcal{T}(\varepsilon)[f(\varepsilon - \mu_\mathrm{L}) - f(\varepsilon - \mu_\mathrm{R})]\,, \tag{4}$$

by $\mathcal{P}_\mathrm{L} = \mu_\mathrm{L} I/e - \mathcal{J}$ and $\mathcal{P}_\mathrm{R} = \mathcal{J} - \mu_\mathrm{R} I/e$.

We are particularly interested in the asymmetry of the dissipation, given by

$$\mathcal{P}_\mathrm{A} = \mathcal{P}_\mathrm{R} - \mathcal{P}_\mathrm{L} = 2\left(\mathcal{J} - \frac{\bar{\mu} I}{e}\right) = \frac{4}{h} \int_{-\infty}^{\infty} \mathrm{d}\varepsilon\, (\varepsilon - \bar{\mu})\mathcal{T}(\varepsilon)[f(\varepsilon - \mu_\mathrm{L}) - f(\varepsilon - \mu_\mathrm{R})]\,. \tag{5}$$

At zero temperature, the difference of Fermi factors limits the energy integration to the interval $[\mu_\mathrm{R}, \mu_\mathrm{L}]$, and therefore the asymmetry $\mathcal{P}_\mathrm{A}$ follows from the energy dependence of $\mathcal{T}(\varepsilon)$ therein. In particular, an approximately energy-independent $\mathcal{T}(\varepsilon)$ in the mentioned interval leads to an almost negligible $\mathcal{P}_\mathrm{A}$, while the fact that $\mathcal{T}(\varepsilon)$ is in most generic situations an increasing function of $\varepsilon$ translates into $\mathcal{P}_\mathrm{A} > 0$ in the case where $\mu_\mathrm{L} > \mu_\mathrm{R}$.

## 2.2 Low-voltage expansion

A quantitative analysis of the lowest order contributions in the voltage $eV = \mu_\mathrm{L} - \mu_\mathrm{R}$ is obtained through an expansion of the transmission

$$\mathcal{T}(\varepsilon) = \mathcal{T}(\bar{\mu}) + (\varepsilon - \bar{\mu})\mathcal{T}'(\bar{\mu}) + V\mathcal{T}'_V(\bar{\mu}) + \mathcal{O}[(\varepsilon - \bar{\mu})^2] + \mathcal{O}[V^2] + \mathcal{O}[V(\varepsilon - \bar{\mu})]\,, \tag{6}$$

where $\mathcal{T}'(\varepsilon)$ denotes the energy-derivative and $\mathcal{T}'_V(\varepsilon)$ the voltage-derivative of the transmission function. Inserting Eq. (6) into the expressions (2) and (3) of the dissipated power at zero temperature yields the lowest order contribution

$$\mathcal{P}_\mathrm{L/R} = \frac{1}{h}\mathcal{T}(\bar{\mu})(eV)^2 + \mathcal{O}[(eV)^3]\,, \tag{7}$$

which is of second order in the bias voltage, and the same on both sides of the scatterer. Of course, the sum of the two contributions reproduces the total dissipated power $\mathcal{P}_\mathrm{T} = (2e^2/h)\,\mathcal{T}(\bar{\mu})V^2 = gV^2 = VI$. The difference of the dissipated powers on the two sides of the scatterer appears only in third order in $V$ as

$$\mathcal{P}_\mathrm{A} = \frac{1}{3h}\mathcal{T}'(\bar{\mu})(eV)^3 + \mathcal{O}[(eV)^4]\,. \tag{8}$$

While the leading-order term of the dissipated power is determined by the transmission at the mean electrochemical potential $\mathcal{T}(\bar{\mu})$, and thus proportional to the conductance, the dominant term of the asymmetry $\mathcal{P}_\mathrm{A}$ is proportional to the energy derivative of the transmission $\mathcal{T}'(\bar{\mu})$. The first-order bias-voltage dependence of the transmission can lead to an electric asymmetry with a current voltage characteristic for which $I(-V) \neq -I(V)$. However, it does not affect the leading order terms in the above expansions. It leads to a third-order term in the power (7) and a fourth-order term in the asymmetry (8). For the particular situation of a structure with left-right symmetry, there is no electric asymmetry, one has $I(-V) = -I(V)$, and the transmission coefficient at a given energy must be an even function of the bias voltage, such that $\mathcal{T}'_V(\bar{\mu}) = 0$. In this case, the second-order corrections to the transmission, that are not written explicitly in Eq. (6), lead only to fourth-order terms in the power (7) and to fifth-order terms in the asymmetry (8).

## 2.3 Low-temperature expansion

In the expansion of the dissipated power for small voltage (see Sec. 2.2), zero temperature was assumed. We now consider the effect of a small temperature $\Delta\mathcal{P}_{\mathrm{L/R}}(T) = \mathcal{P}_{\mathrm{L/R}}(T) - \mathcal{P}_{\mathrm{L/R}}(0)$ on the dissipated powers $\mathcal{P}_{\mathrm{L/R}}(T)$ at finite voltage. The effect of a nonzero temperature is that of allowing for the occupation of states at energies that are of order $k_{\mathrm{B}}T$ above the electrochemical potentials, while the occupation of the states just below is reduced. Transmission processes outside the energy window $[\mu_{\mathrm{R}}, \mu_{\mathrm{L}}]$ by about $k_{\mathrm{B}}T$ appear, while those inside the interval close to its edges are reduced, leading to modifications of the asymmetry in the dissipated powers in the presence of an energy-dependent transmission probability.

In order to get a quantitative estimate of $\Delta\mathcal{P}_{\mathrm{L/R}}(T)$, we start from the general expressions (2) and (3). The only temperature dependence is in the Fermi-Dirac distribution functions, and we can write

$$\Delta\mathcal{P}_{\mathrm{L}}(T) = \frac{2}{h}\int_{-\infty}^{\infty} d\varepsilon\, (\mu_{\mathrm{L}} - \varepsilon)\mathcal{T}(\varepsilon)[\Delta f(\varepsilon - \mu_{\mathrm{L}}) - \Delta f(\varepsilon - \mu_{\mathrm{R}})], \tag{9a}$$

$$\Delta\mathcal{P}_{\mathrm{R}}(T) = \frac{2}{h}\int_{-\infty}^{\infty} d\varepsilon\, (\varepsilon - \mu_{\mathrm{R}})\mathcal{T}(\varepsilon)[\Delta f(\varepsilon - \mu_{\mathrm{L}}) - \Delta f(\varepsilon - \mu_{\mathrm{R}})], \tag{9b}$$

where we have defined the temperature-induced change of the Fermi-Dirac distribution $\Delta f(x) = f(x) - \theta(-x)$, where $\theta(-x)$ is the zero-temperature distribution given in terms of the Heaviside step function. This change $\Delta f(x)$ is significant around $x = 0$ and exponentially suppressed on a scale of $k_{\mathrm{B}}T$ when $|x|$ increases. Therefore, at low temperature, only energies in the vicinity of the electrochemical potentials $\mu_{\mathrm{L}}$ and $\mu_{\mathrm{R}}$ contribute to the integrals in Eqs. (9). We then treat the two regions separately in a Sommerfeld expansion approach, and use Taylor expansions of the transmission around those energies

$$\mathcal{T}(\varepsilon) = \sum_{n=0}^{\infty} \frac{\mathcal{T}^{(n)}(\mu_{\mathrm{L/R}})}{n!}(\varepsilon - \mu_{\mathrm{L/R}})^n, \tag{10}$$

where $\mathcal{T}^{(n)}$ is the $n^{\mathrm{th}}$ derivative of the transmission with respect to energy. For the dissipated power on the left, this results in

$$\Delta\mathcal{P}_{\mathrm{L}}(T) = -\frac{2}{h}\sum_{n=0}^{\infty}\frac{\mathcal{T}^{(n)}(\mu_{\mathrm{L}})}{n!}\mathcal{I}_{n+1} + \frac{2}{h}\sum_{n=0}^{\infty}\frac{\mathcal{T}^{(n)}(\mu_{\mathrm{R}})}{n!}(\mathcal{I}_{n+1} - eV\mathcal{I}_n), \tag{11}$$

where we have defined the integrals

$$\mathcal{I}_n = \int_{-\infty}^{\infty} dx\, x^n \Delta f(x). \tag{12}$$

Since $\Delta f(x)$ is an odd function, $\mathcal{I}_n = 0$ for all even $n$. For odd $n$, one has

$$\mathcal{I}_n = 2(k_{\mathrm{B}}T)^{n+1}\int_{0}^{\infty} dx\, \frac{x^n}{1 + e^x} = (2\pi k_{\mathrm{B}}T)^{n+1}\frac{(1 - 2^{-n})|B_{n+1}|}{n+1}, \tag{13}$$

where $B_{n+1}$ are the Bernoulli numbers. The expansion of the transmission around the electrochemical potential results in an expansion in powers of the temperature. The lowest order contribution is due to the terms involving $\mathcal{I}_1 = (\pi^2/6)(k_{\mathrm{B}}T)^2$. Collecting all those terms, we have the lowest order temperature correction to the dissipated power in the left electrode

$$\Delta\mathcal{P}_{\mathrm{L}}(T) = -\frac{\pi^2}{3h}\big[\mathcal{T}(\mu_{\mathrm{L}}) - \mathcal{T}(\mu_{\mathrm{R}}) + eV\mathcal{T}'(\mu_{\mathrm{R}})\big](k_{\mathrm{B}}T)^2 + \mathcal{O}[(k_{\mathrm{B}}T)^4]. \tag{14}$$

An analogous treatment of the power on the right side yields

$$\Delta\mathcal{P}_{\mathrm{R}}(T) = \frac{2}{h}\sum_{n=0}^{\infty}\frac{\mathcal{T}^{(n)}(\mu_{\mathrm{L}})}{n!}(\mathcal{I}_{n+1} + eV\mathcal{I}_n) - \frac{2}{h}\sum_{n=0}^{\infty}\frac{\mathcal{T}^{(n)}(\mu_{\mathrm{R}})}{n!}\mathcal{I}_{n+1}\,, \tag{15}$$

and the lowest order correction at low temperature

$$\Delta\mathcal{P}_{\mathrm{R}}(T) = \frac{\pi^2}{3h}\left[\mathcal{T}(\mu_{\mathrm{L}}) - \mathcal{T}(\mu_{\mathrm{R}}) + eV\mathcal{T}'(\mu_{\mathrm{L}})\right](k_{\mathrm{B}}T)^2 + \mathcal{O}[(k_{\mathrm{B}}T)^4]\,. \tag{16}$$

It then appears that for a scatterer with constant transmission, the temperature does not affect the dissipated power, at least in lowest order. In contrast, in a situation where the transmission $\mathcal{T}(\varepsilon)$ increases with energy, the correction on the left $\Delta\mathcal{P}_{\mathrm{L}}(T)$ is negative while $\Delta\mathcal{P}_{\mathrm{R}}(T)$ increases with temperature. As a consequence, the asymmetry also increases with temperature, consistent with the result of Ref. [17], and one can even imagine situations where the dissipated power in the left electrode becomes negative such that a cooling of that electrode occurs [27].

A strong temperature effect can be expected when a conductance step occurs at an energy between $\mu_{\mathrm{R}}$ and $\mu_{\mathrm{L}}$, such that $\mathcal{T}(\mu_{\mathrm{L}}) - \mathcal{T}(\mu_{\mathrm{R}}) = 1$ and $\mathcal{T}'(\mu_{\mathrm{L}}) = \mathcal{T}'(\mu_{\mathrm{R}}) = 0$. Then, one gets the simple result

$$\Delta\mathcal{P}_{\mathrm{L/R}}(T) = \mp\frac{\pi^2}{3h}(k_{\mathrm{B}}T)^2 + \mathcal{O}[(k_{\mathrm{B}}T)^4]\,, \tag{17}$$

which is independent of the bias voltage.

## 3 Asymmetric dissipation around a QPC

Among the usual scatterers considered in the mesoscopic regime, a QPC is particularly interesting since at the conductance plateaus $\mathcal{T}'(\bar{\mu}) = 0$, which leads, according to the formalism developed above, to important consequences on the features of the power dissipation.

### 3.1 Transmission of an abrupt QPC

The most prominent feature of a QPC is the observed conductance quantization at integer multiples of $2e^2/h$ [28, 29]. The robustness of such a behavior allows for different theoretical descriptions that result in transmission coefficients $\mathcal{T}(\varepsilon)$ exhibiting, as a function of $\varepsilon$, extended plateaus separated by fast ascents. Among them, there exists the adiabatic approximation applicable to a smooth constriction [30], the exact treatment of a double-harmonic-oscillator saddle-point potential [31], and the wavefunction matching for an abrupt constriction [32]. We adopt the latter description, considering a narrow channel of length $L$ and width $2w$ connecting two wide regions of width $2W$, with $W \gg w$ [see the inset of Fig. 2(a)].

A convenient way of implementing the approach of Ref. [32] is to use the one-to-one correspondence between the quantized channels within the constriction and the transmission eigenmodes of the scatterer, defined as the eigenvectors of the $N_{\mathrm{L}} \times N_{\mathrm{L}}$ matrix $t^{\dagger}t$, and labeled by the positive integer $n$ [9, 23]. The quantized channels within the constriction are defined by a transverse wavevector $Q_n = \pi n/2w$ and a quantized transverse energy $E_n = \hbar^2 Q_n^2/2M_{\mathrm{e}}$ (we note $M_{\mathrm{e}}$ the effective electron mass). The associated longitudinal wavevector $K_n = \left(k^2 - Q_n^2\right)^{1/2}$ is real for the open (conducting) channels with $E_n \leq \varepsilon$, and pure imaginary for the closed (evanescent) channels with $E_n \geq \varepsilon$ (the wavevector $k$ is defined by $\varepsilon = \hbar^2 k^2/2M_{\mathrm{e}}$). The transmission coefficient is $\mathcal{T}(\varepsilon) = \sum_n^{N_{\mathrm{L}}} \mathcal{T}_n(\varepsilon)$, where $\mathcal{T}_n(\varepsilon)$ is the $n^{\mathrm{th}}$ transmission eigenvalue.

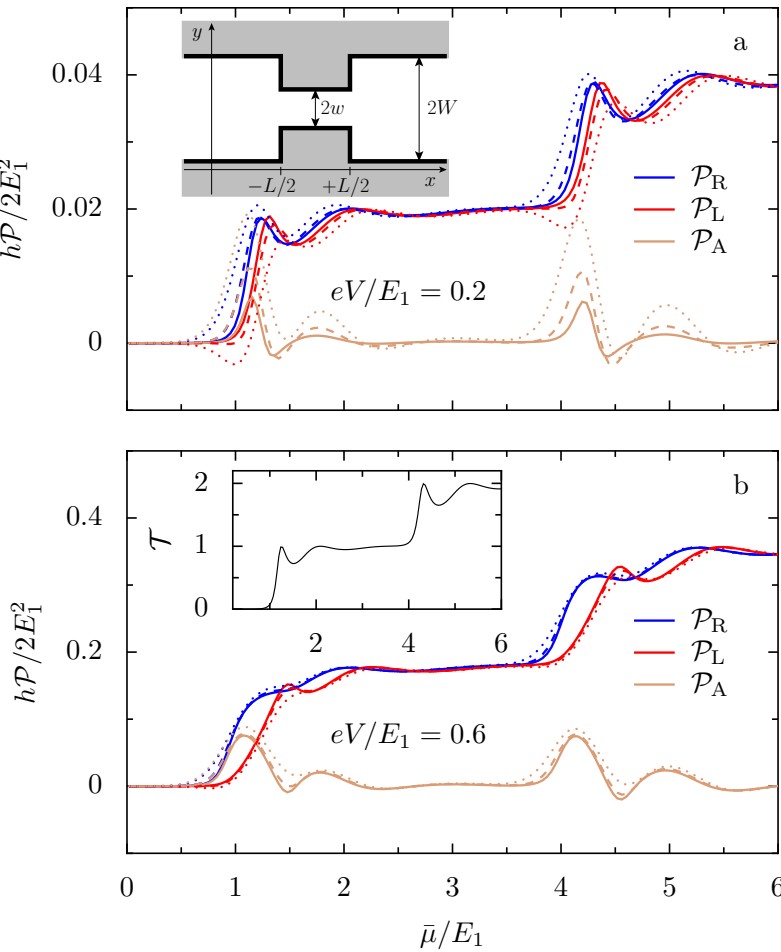

Figure 2: Dissipated power as a function of the mean electrochemical potential $\bar{\mu}$ at the right ($\mathcal{P}_{\mathrm{R}}$, blue) and left ($\mathcal{P}_{\mathrm{L}}$, red) of an abrupt constriction (sketched in the inset of panel a), together with their difference ($\mathcal{P}_{\mathrm{A}}$, ocher) for low bias voltages, progressively departing from the linear regime [(a) and (b)]. Solid lines are for zero temperature, dashed and dotted lines are results for temperatures with $k_{\mathrm{B}}T/E_1 = 0.04$ and 0.08, respectively. The chosen energy scale $E_1$ corresponds to that of the first transverse mode in the narrow part, which for a QPC in a GaAs/AlGaAs heterostructure is given by $E_1 \approx 5.6\,\mathrm{eVnm}^2/(2w)^2$. Inset of panel b: transmission coefficient as a function of $\bar{\mu}/E_1$.

As remarked in Ref. [32], the channels of the wide region that are not mismatched with the $n^{\mathrm{th}}$ channel of the constriction belong to the interval $\Delta Q_n = [Q_{n-1}, Q_{n+1}]$. Restricting ourselves to the previous interval, we obtain an average generalized longitudinal wavevector

$$\mathcal{K}_n = \frac{w}{\pi} \int_{\Delta Q_n} \mathrm{d}q \, \sqrt{k^2 - q^2} \,. \tag{18}$$

According to the positioning of $k$ with respect to the integration interval, $\mathcal{K}_n$ may have real and/or imaginary parts. The transmission eigenvalue associated with the channel $n$ is given by $\mathcal{T}_n(\varepsilon) = \tau_n^2(\varepsilon)$, with [9]

$$\tau_n(\varepsilon) = \frac{4|K_n|\,\mathrm{Re}\{\mathcal{K}_n\}}{|D_n|} \,, \tag{19}$$

and

$$D_n = (K_n + \mathcal{K}_n)^2 \, \mathrm{e}^{-\mathrm{i}K_n L} - (K_n - \mathcal{K}_n)^2 \, \mathrm{e}^{\mathrm{i}K_n L} \,. \tag{20}$$

The zero-temperature linear conductance resulting from Eq. (19) [shown in the inset of Fig. 2(b)] provides a very good approximation to the numerical quantum results [9, 32]. The overall increase of the conductance by plateaus as a function of $\varepsilon$ coexists with oscillations resulting from quantum interference within the abrupt constriction. This resonant behavior is suppressed when considering more realistic smoother constrictions, as well as by the effect of finite temperature [33].

The previous approach can be extended in order to incorporate the effect of a finite bias on the transmission coefficient by using the ansatz of a linear potential drop that occurs within the QPC [23]. However, as discussed in Sec. 2, we can ignore the resulting corrections if we restrict ourselves to relatively small bias.

## 3.2 Dissipated power around an abrupt QPC

In Fig. 2 we present the dissipated power at the right and left of the scatterer ($\mathcal{P}_R$ in blue and $\mathcal{P}_L$ in red), together with their difference ($\mathcal{P}_A$ in ocher) for the cases of a low and high bias voltage [Figs. 2(a) and 2(b), respectively]. The solid lines represent the zero temperature result, dashed and dotted lines are for increasing finite temperatures, resulting from the application of Eq. (2) with Eqs. (3), (5), and (19), for the abrupt constriction sketched in the inset of Fig. 2(a). To present data in the figures, we use as energy scale $E_1 = \hbar^2 \pi^2 / 8 M_e w^2$, which is the lowest transverse quantized energy in the narrow part of the system. For the case of a two-dimensional electron gas in a GaAs/AlGaAs heterostructure, this energy is $E_1 \approx 5.6\,\mathrm{eVnm}^2/(2w)^2$. For a QPC of width $2w = 40\,\mathrm{nm}$ one therefore has $E_1 \approx 3.5\,\mathrm{meV}$, and a temperature of $k_B T/E_1 = 0.1$ corresponds to $T \approx 4\,\mathrm{K}$.

### 3.2.1 Zero-temperature behavior of the dissipation

At low bias voltage, $\mathcal{P}_R$ and $\mathcal{P}_L$ follow the increase of the transmission coefficient [shown in the inset of Fig. 2(b)] as a function of the Fermi energy $\bar{\mu}$, as expected from the low-voltage expansion of Eq. (7). The power dissipation asymmetry $\mathcal{P}_A$ is considerably reduced in the regions where the mean electrochemical potential exhibits conductance plateaus, as a consequence of the approximate symmetry with respect to $\bar{\mu}$ of the integrand in Eq. (5). As expected from Eq. (8), $\mathcal{P}_A$ follows the energy-derivative of the transmission curve. Therefore, the power dissipation on both sides is determined by the transmission of electrons through the QPC, with each channel contributing to the dissipated power. The asymmetry in the power dissipation $\mathcal{P}_A$ is most pronounced close to the conductance steps, where the opening of a new channel leads to a large energy-dependence of the transmission. An increasing bias voltage leads to a smoothing of the previous structure, with an increase of $\mathcal{P}_R$ and $\mathcal{P}_L$, and also an increase of $\mathcal{P}_A$ that is consistent with the low-voltage limits of Eqs. (7) and (8). The oscillations of the transmission on the conductance plateaus [see the inset of Fig. 2(b)] are a consequence of the abrupt shape of the QPC. Since the power dissipation asymmetry follows the energy-derivative of the transmission, a negative dissipation asymmetry appears at mean electrochemical potential values where the transmission decreases. That those features are observed in Fig. 2 for an abrupt QPC confirms the general validity of our low-voltage expansions in Sec. 2.2. However, such conductance oscillations and points with negative power asymmetry do not occur in adiabatic QPC models.

The increase of $\mathcal{P}_R$ with $\bar{\mu}$ and with the bias voltage $V$ is put in evidence in Fig. 3(a), where the dissipated power in the right lead is shown in colorscale as a function of the bias voltage and the mean electrochemical potential. The dissipated power increases with increasing bias, with contributions of the two conductance channels appearing at the energies expected from the conductance steps [see the inset of Fig. 2(a)]. The asymmetry in dissipated power $\mathcal{P}_A$ is shown in Fig. 3(b). For the values of $\bar{\mu}$ corresponding to a conductance plateau $\mathcal{T}_n(\varepsilon) \simeq 1$

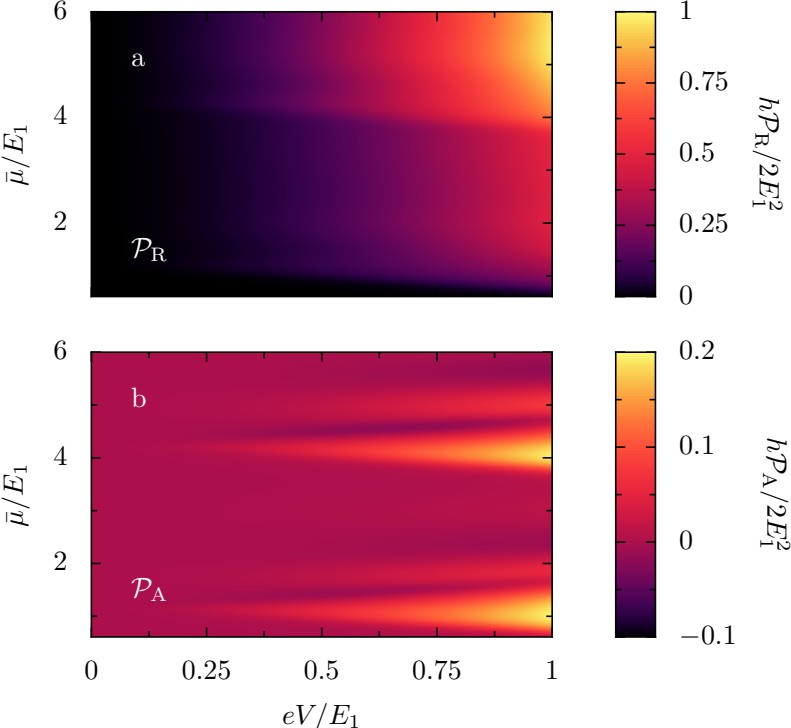

Figure 3: Colorscale plot of the dissipated power on the right $\mathcal{P}_R$ (panel a) and the difference $\mathcal{P}_A$ (panel b), as a function of the applied voltage and the mean electrochemical potential, at zero temperature.

and $\mathcal{P}_A$ is close to zero, while on the conductance steps $\mathcal{P}_A$ strongly increases with the bias voltage. Horizontal cuts along the voltage axis in Fig. 3 confirm that the increase of $\mathcal{P}_R$ starts proportional to the square of the voltage as expected from Eq. (7), while $\mathcal{P}_A$ starts proportional to $(eV)^3$ at low voltage as predicted from Eq. (8). At very large voltages, the results should be taken with care since our model does not include electron-electron interactions. It cannot be excluded that they influence the transmission and thus also the power dissipation in the regime of strong bias voltage.

On a larger scale of electrochemical potentials, the dissipated power follows an approximate law as $\mathcal{P}_R \propto \bar{\mu}^{1/2}$. Such a behavior can be traced back to the plateau widening as we increase the Fermi energy (since $E_n \propto n^2$, the plateau extent verifies $\Delta E_n \propto n \propto \sqrt{E_n}$). The dependence of the positions of the conductance steps $E_n \propto n^2$ translates into $\mathcal{T} \approx n \propto \bar{\mu}^{1/2}$ and then, since according to Eq. (7), at low voltage $\mathcal{P}_{R/L} \propto \mathcal{T}$ the dissipated power increases with the square root of $\bar{\mu}$. In contrast, the asymmetry of the power dissipation $\mathcal{P}_A$ is proportional to the energy-derivative of the transmission. Since the conductance steps are all of the same height, and the steepness does not increase with energy, the asymmetry in the subsequent steps does not increase with $\bar{\mu}$, such that the power difference becomes negligible as compared to the individual values of dissipated power $\mathcal{P}_{R/L}$ when the electrochemical potentials are large and many conductance channels are open.

### 3.2.2 Increase of asymmetry with temperature

As expected from the lowest order terms of the low-temperature expansion of the dissipated powers presented in Sec. 2.3, the maximum values of the asymmetry increase at finite temperature (dashed and dotted lines in Fig. 2). In Fig. 4 the full temperature dependence of the dissipated power on both sides of the QPC together with the asymmetry (ocher lines) for the

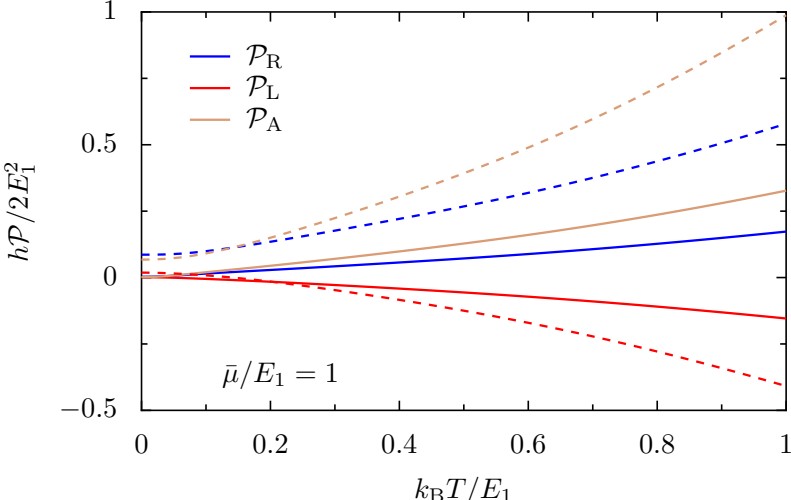

Figure 4: Dissipated power on the right $\mathcal{P}_R$ (blue), on the left $\mathcal{P}_L$ (red), and the asymmetry $\mathcal{P}_A$ (ocher) as a function of the temperature, for a mean electrochemical potential of $\bar{\mu}/E_1 = 1$ situated in the first conductance step. Solid and dashed lines are for voltages of $eV/E_1 = 0.2$ and $0.6$, respectively.

mean electrochemical potential tuned close to the first conductance step is shown. It appears that the increase of the asymmetry continues far beyond the validity of the lowest-order term (16) of the expansion, up to large values of the temperature with $k_B T$ much larger than $eV$. The reason for this temperature-induced increase is the possibility to transmit electrons at high energy, even above the upper (left) chemical potential. These processes remove high-energy electrons from the left reservoir, and contribute a negative power dissipation there [27]. On the right side of the QPC, those electrons lead to a particularly high power dissipation due to the large amount of energy to dissipate. When the upper chemical potential is placed close to a conductance step, these effects can be important due to the large increase of transmission with energy, while the processes at energies below the lower (right) chemical potential, that have an opposite effect, are reduced by much lower transmission values.

## 4 Estimate of the position of maximum dissipation

The previous analysis considered the asymmetry of the power dissipation between the two sides of a QPC without providing any spatial resolution. Since preliminary studies [13] were able to observe a hot spot along the path of the electrons after traversing the QPC, it is important to estimate the lengthscale on which the dissipation takes place.

In the formalism described in the last sections we assumed that an electron with energy $\varepsilon$ traverses the constriction elastically, encountering at the right of the QPC a 2DEG with an electrochemical potential $\mu_R$. The excess energy $\varepsilon - \mu_R$ of this hot electron is dissipated through inelastic scattering on the scale of the inelastic mean free path

$$l(\varepsilon) = v(\varepsilon)\tau_i(\varepsilon). \tag{21}$$

In the equation above $v(\varepsilon) = \sqrt{2(\varepsilon + eV/2)/M_e}$ is the electron velocity at the right of the QPC, under the assumption that the whole potential drop occurs in the QPC region (as sketched in Fig. 1) and that the electron motion is ballistic because the small-angle scattering is very weak. In Eq. (21) $\tau_i(\varepsilon)$ stands for the inelastic scattering time (or quasiparticle lifetime) set by the electron-electron and electron-phonon interactions. The excess energy and the electron

density of the 2DEG determine the predominance of one mechanism over the other, and, more generally, whether they can be disentangled or a coupled-mode description is needed.

Sufficiently close to the Fermi energy, Landau quasiparticles have a lifetime limited by electron-electron interactions, which scales inversely to the square of the excess energy in three dimensions, and acquires an additional logarithmic correction for the case of the 2DEG [34,35]. Alternatively, we can write $\tau_i(\varepsilon) = \hbar/2\Gamma(\varepsilon)$, and obtain the damping rate $\Gamma(\varepsilon)$ from the imaginary part of the quasiparticle self-energy. For the latter the random phase approximation can be implemented, treating the electron-electron and electron-phonon couplings on the same footing [36]. While such an approach needs to be numerically implemented, it has the advantage of not being restricted to small excess energies, allowing to incorporate the effect of the finite thickness of the 2DEG, and considering different kinds of phonons (i.e., acoustic *versus* optical, bulk *versus* interface) [37,38]. The quasiparticles can be scattered either by the excitation of electron-hole pairs or by the emission of a coupled plasmon-phonon mode. For excess energies below the threshold of the latter mechanism, the damping rate scales approximately as [36,37,39]

$$\frac{\Gamma(\varepsilon)}{\varepsilon_F} = a\left(\frac{\varepsilon}{\varepsilon_F} - 1\right)^2, \tag{22}$$

where the dimensionless constant $a$ is weakly dependent on the electron density of the 2DEG.

Putting together Eqs. (21) and (22) for the hot electron arriving in the 2DEG at the right side of the QPC, we have

$$l(\varepsilon) = \frac{b(\varepsilon + eV/2)^{1/2}}{(\varepsilon - \mu_R)^2}, \tag{23}$$

with $b = \hbar\mu_R/a\sqrt{2M_e}$.

Assuming that each hot electron with energy $\varepsilon$ releases all its excess energy precisely at a distance $l(\varepsilon)$ from the QPC, we define the power dissipated per unit length as

$$\mathfrak{p}_R(r) = \int_{-\infty}^{\infty} d\varepsilon\, p_R(\varepsilon)\delta(l(\varepsilon) - r), \tag{24}$$

where $p_R(\varepsilon)$ is given in Eq. (3b), that verifies $\mathcal{P}_R = \int_0^{\infty} dr\, \mathfrak{p}_R(r)$ and therefore simply introduces a change of variables in Eq. (2). We thus have

$$\mathfrak{p}_R(r) = \left.\frac{p_R(\varepsilon)}{|l'(\varepsilon)|}\right|_{\varepsilon = l^{-1}(r)}, \tag{25}$$

where $l'$ is the $\varepsilon$-derivative and $l^{-1}$ the inverse of the function $l$ defined in Eq. (21). We do not aim to resolve the angular dependence of the dissipation, and thus $r$ represents the radial distance from the QPC.

Since $l'(\varepsilon) < 0$, the position of maximum dissipation will be given by setting

$$\frac{d\mathfrak{p}_R(r)}{dr} = -p'_R\left(l^{-1}(r)\right)\frac{dl^{-1}}{dr} - p_R\left(l^{-1}(r)\right)\frac{d^2l^{-1}}{dr^2} = 0. \tag{26}$$

Given the difficulty to invert the function $l(\varepsilon)$, it is convenient to express the condition (26) in terms of the variable $\varepsilon$ as

$$\begin{aligned}\left[24(\varepsilon + eV/2)^2 - 8(\varepsilon + eV/2)(\varepsilon - \mu_R) - (\varepsilon - \mu_R)^2\right]p_R(\varepsilon) \\ + 2(\varepsilon + eV/2)(\varepsilon - \mu_R)\left[4(\varepsilon + eV/2) - (\varepsilon - \mu_R)\right]p'_R(\varepsilon) = 0.\end{aligned}$$

The numerical solution of this transcendental equation yields a value $\varepsilon^*$, and the point of maximum dissipation $r^* = l(\varepsilon^*)$. While $\varepsilon^*$ does not depend on the parameters $a$ and $b$, $r^*$

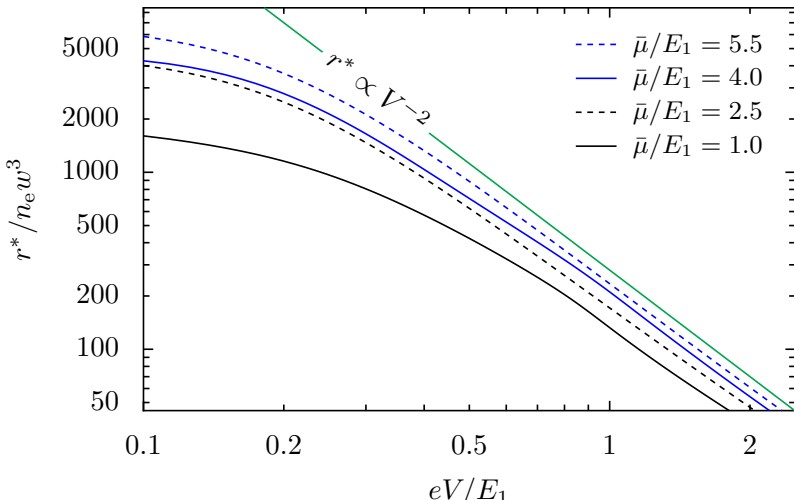

Figure 5: Estimate of the distance $r^*$ at the right of the QPC where the maximum power dissipation $\mathcal{P}_R$ occurs, as a function of the bias voltage $V$. The distances are given in units of $n_e w^3$, where $n_e$ is the electron density of the 2DEG and $2w$ is the width of the QPC, while the energies are scaled with the one of the first transverse mode of the QPC. Solid lines are for values of $\bar{\mu}$ in a conductance step, dashed lines on plateaus. The green line indicates the slope of a power law $r^* \propto V^{-2}$. The temperature is given by $k_B T / E_1 = 0.04$.

does. For electron densities of $2 \times 10^{11}\,\mathrm{cm}^{-2}$ and $10^{12}\,\mathrm{cm}^{-2}$ the values of $a$ are roughly equal to 0.04 and the variation within such an interval is weak [36, 37]. Since the typical electron densities in this kind of experiments are between $10^{10}$ and $10^{12}\,\mathrm{cm}^{-2}$ [40], we adopt the previous value of $a$, and provide in Fig. 5 the results for $r^*$ as a function of the bias voltage, for various values of $\bar{\mu}$. The general trend is a decrease of $r^*$ with increasing bias, which is readily explained since the high energy of the hot electron leads to a large dissipation energy with a short lifetime. In addition, $r^*$ increases with $\bar{\mu}$, an effect that is related to the increased velocity of the hot electrons. At large voltage, the voltage dependence of $r^*$ approaches the power law $r^* \propto V^{-2}$ indicated by the green line. For the second conductance step of a QPC at $\bar{\mu} = 4E_1$ in a 2DEG with density $2 \times 10^{11}\,\mathrm{cm}^{-2}$, and a typical value $V \simeq 4\,\mathrm{mV}$ of the bias voltage, we obtain $r^* \simeq 2\,\mu\mathrm{m}$, in line with the order of magnitude of the experimental findings of Ref. [13].

It is important to stress that the previous analysis is valid for bias voltages which are much smaller than $\bar{\mu}$ since we assumed the form (22) of the damping rate where the excitation of electron-hole pairs constitutes the first step towards the thermalization within the 2DEG and which is valid for excess energies below the threshold for the emission of plasmon-phonon modes. For a density $n_e = 2 \times 10^{11}\,\mathrm{cm}^{-2}$, where the plasmon-phonon coupling is weak, such a threshold happens for $|V| = 12\,\mathrm{mV}$, while for $n_e = 10^{12}\,\mathrm{cm}^{-2}$, where the plasmon-phonon coupling is strong, the threshold is at $|V| = 10\,\mathrm{mV}$ [36, 37]. Once this channel is opened, a dramatic increase of the relaxation rate is associated which is expected to lead to a strong reduction of $r^*$.

The present analysis yields an estimation of the position of maximum dissipation, but does not give any information about the extent of the hot spot. The modelization of this experimentally relevant parameter would require to complement the approach by describing the detail of the energy flow from the 2DEG to the ionic lattice.

## 5   Conclusions

Motivated by spatially resolved nanothermometry measurements in the region of the current flow through a QPC [13], we have investigated the dissipated power based on a Landauer-Büttiker quantum transport approach. We have shown that an asymmetry between the power dissipated on the two sides of the QPC occurs when the transmission of the QPC depends on energy. For the generic case of a transmission that increases with energy, the dissipated power is higher on the side of the QPC located downstream with respect to the electron flow. The asymmetry is most pronounced close to the conductance steps of the QPC where this energy dependence is particularly strong, while it is weak on the conductance plateaus. A temperature expansion indicates that the asymmetry is enhanced by increasing temperature at low temperatures.

For the example of an abrupt QPC, we have used the known result [9,32] for the energy dependent transmission of that model to calculate explicitly the dependence of the dissipated power and the asymmetry on the mean electrochemical potential, the voltage, and the temperature. The results confirm the general considerations of Sec. 2, whose qualitative features are independent of the details of the QPC modeling. They indicate that when the QPC is tuned to the first step, one can reach at finite temperature a regime where the power on the upstream side becomes negative, such that a cooling effect occurs, consistent with the predictions of Ref. [27].

We have estimated the distance from the QPC to the hot spot where the highest power dissipation per unit length occurs, based on the relaxation rate of quasiparticle excitations in a 2DEG of Refs. [36, 37]. The distance of such an expected hot spot from the QPC decreases strongly with the applied voltage. Using typical values for experiments, we got an order of magnitude that is consistent with the preliminary nanothermometry measurements of Ref. [13].

It will be highly desirable to extend our work towards a theory of the dissipated power in quantum transport devices with full spatial resolution, with an improved understanding of the relation between nonlocal quantum transport properties and local energy dissipation.

While we have concentrated our analysis on the experimentally relevant case of a QPC defined in a 2DEG, the overall conclusion of an asymmetric power dissipation in cases with a strong energy-dependence of the transmission coefficient is quite general to quantum electronic transport, and it can be applied to other setups like atomic and molecular junctions [15]. In particular, it can be useful to understand the structural fluctuations of an atomic-scale junction in a low-temperature scanning tunneling microscope [41]. The dependence of the irreversible changes in the properties of the junction on the current direction [42] could have as origin the asymmetric dissipation that we have characterized in this work.

## Acknowledgements

We thank Eli Zeldov for sharing with us unpublished experimental data, as well as Denis Basko, Nico Leumer, Anna Rosławska, and Robert Whitney for useful discussions.

**Funding information**   Financial support from the French National Research Agency ANR through project ANR-20-CE30-0028-01 is gratefully acknowledged. This work of the Interdisciplinary Thematic Institute QMat, as part of the ITI 2021-2028 program of the University of Strasbourg, CNRS, and Inserm, was supported by IdEx Unistra (ANR 10 IDEX 0002), and by SFRI STRAT'US project (ANR 20 SFRI 0012) and EUR QMAT ANR-17-EURE-0024 under the framework of the French Investments for the Future Program.

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
