# Peer review of "Asymmetric power dissipation in electronic transport through a quantum point contact"

_SciPost Physics, doi:SciPost Phys. 12, 105 (2022)_

## Round 3 · Referee Report · David Sanchez (Referee 1) · 2022-1-21

Strengths

  1. The topic is well motivated.
  2. The model and its assumptions are clearly discussed.
  3. The results are relevant in view of recent experimental developments.

Weaknesses

  1. The model is restricted to left-right symmetric devices.
  2. The model considers noninteracting electrons only but in some cases large biases. Its general validity is therefore doubtful.

Report

The manuscript investigates possible asymmetric dissipations in quantum point contacts. The asymmetry manifests itself in different heat currents measured at the two reservoirs attached to a point contact. The topic is timely, the paper reads well and the results are discussed in detail. I recommend the paper for publication with minor corrections.

Requested changes

  1. The authors should comment how much their conclusions would be affected by left-right symmetry breakings. This is important because 1) no device will be fully symmetric and 2) the discussed power asymmetries might be a tool to quantify the left-right asymmetry present in the system, which is not possible with the charge current.

  2. Refs. 15-17 distinguish between contact asymmetry (the one considered here) and electric asymmetry. For completeness, the authors should also discuss the latter, even in brief terms. I think this would be beneficial for the reader.

  3. Experimental QPCs are better described with adiabatic confinement potentials. Here, the authors prefer abrupt interfaces, which cause unwanted plateau oscillations. I think the simulations would be cleaner with smooth potentials. I am not sure why the authors choose hard wall potentials. If it is due to simplicity, they should say so.

  4. The voltage issue is trickier. The authors restrict themselves to low voltages but in Fig. 3 they show results for up to 3.5 mV, which is large for a mesoscopic device. Either the authors plot results for voltages much smaller than temperature or they alert the reader that at high voltage their results are only qualitative and will likely change in the presence of interactions.

  • validity: good
  • significance: high
  • originality: ok
  • clarity: high
  • formatting: perfect
  • grammar: perfect

Author:  Dietmar Weinmann  on 2022-02-15  [id 2204]

(in reply to Report 1 by David Sanchez on 2022-01-21)

We thank the referee for his positive overall judgment and useful remarks. We have taken into account the suggested changes as explained in detail below.

  1. Our model and our results are not limited to left-right symmetric systems since the equality between transmission coefficients for both directions of carrier transport does not require such a symmetry. It was only when presenting the low-voltage expansion of Sec. 2.2 and in a sentence that was placed in the 2nd paragraph of Sec. 2 in the original version that we adopted this simplifying hypothesis, which might have triggered the referee's comment. Indeed, the lowest order correction to the transmission coefficient is of second order in the bias voltage for left-right symmetric systems, but a first-order term cannot be excluded in the absence of such a symmetry. Following the remark of the referee, and in order to clarify the generality of our results, we have included in the revised version a first-order term in the low-voltage expansions in Sec. 2.2 and shown that it does not affect the leading order terms for the power dissipation and its asymmetry, which scales with the third power of the bias voltage. The results presented in Sec. 2 are therefore also valid for systems without left-right symmetry. While the example of the system chosen to obtain quantitative results in Sec. 3 has left-right symmetry, we have demonstrated that there is no strong influence of such a symmetry on the power dissipation asymmetry at low bias voltage. To make this point clear, in the revised version of our paper we have explicitly included a first-order term in bias in the expansion of the transmission (6), and we have introduced a discussion of its consequences for the expansions (7) and (8) of the dissipated power and the asymmetry below Eq. (8). Moreover, we have moved the comment about the absence of a first-order correction in bias voltage to the transmission for the special case of left-right symmetric systems from the paragraph above Sec. 2.1 to the end of Sec. 2.2.
  2. As mentioned above, our results are not restricted to the case of geometrical or electrical symmetry. In the revised version, we mention the particular case of the electric asymmetry in the discussion of the low-voltage expansion at the end of Sec. 2.2.
  3. For the example model of Sec. 3, we have chosen an abrupt QPC since it is a 2d model with a low number of parameters and for which an analytic solution for the energy-dependent transmission coefficient is available. We agree with the referee that the plateau oscillations would not appear for adiabatic QPC models with smooth width variation or smooth saddle-point shaped potential. We included a discussion of the issue at the end of the first paragraph in Sec. 3.2.1. indicating which are the specific features of Fig. 2 linked with the abrupt character of of the QPC.
  4. The discussion of the effect of electron-electron interaction and the possible implementation of a self-consistent treatment were already presented in the second paragraph of Sec. 2, but not in Sec. 3, where some plots are extended to relatively large voltage values. Following the referee's suggestion, we added two sentences at the end of the second paragraph of Sec. 3.2.1. (where Fig. 3 is discussed) to alert the reader about the possible influence of electron-electron interactions at the strongest bias therein considered.

---

## Round 3 · Referee Report · Anonymous (Referee 3) · 2022-1-28

Strengths

  1. General consideration which provides qualitatively correct estimate of heating irrespective to specificities of the system.
  2. The consideration is new (in a sense that nobody previously performed such an analysis) and interesting.

Weaknesses

  1. Authors use assumptions (such as choice of potential profile) which while plausible are not the only possibility. This results in conclusions being dependent on these arbitrary assumptions.
  2. Paper is not self-contained: many results are introduced without any explanations simply indicating where they come from (such are, e.g., analysis of transmission in QPC or expression for damping rate scales).

Report

Novelty of the consideration and its general qualitative validity are very attractive. Thus, I do not hesitate to recommend acceptance of the manuscript.

Requested changes

  1. It would be helpful for a general reader if authors try to make the presentation a bit more self-contained.
  2. More detailed discussion of choice of parameters would help. Currently parameters seem to be chosen to make final result coincide with experimental data (which is ok) without discussion of choice for each of the parameters.

  • validity: good
  • significance: high
  • originality: top
  • clarity: good
  • formatting: excellent
  • grammar: -

Author:  Dietmar Weinmann  on 2022-02-15  [id 2205]

(in reply to Report 3 on 2022-01-28)

We thank the referee for his/her positive overall judgment and useful remarks. We have taken into account the suggestions as explained in detail below.

  1. The referee feels that the paper is not self-contained. We indeed use concepts and results from the literature, describing the main lines, and indicating the references upon which we build our analysis. We feel that a more detailed presentation of those aspects would not be useful and might divert the reader from the essential new physics that we present in our work. For example, we have outlined the main lines of the derivation of the transmission probability for an abrupt constriction in Sec. 3.1, but without rewriting the full description of the approach that can be found in Ref. [9]. The case of the damping rate is similar. We have presented a way to estimate the distance of the point with the strongest heat dissipation from the QPC in Sec. 4. from the knowledge of the damping rate in the 2DEG. Such a parameter is extracted from the literature taking into account the typical electron densities and the bias voltages used in the experiment of Ref. [13]. We believe that a detailed description of the evaluation of that parameter is not crucial for the understanding of our approach. Moreover, our approach remains valid in systems where other processes might dominate the damping rate. Of course, for a reader interested in the details of the damping rate in a 2DEG, we provide references [34-39], including a book.
  2. The referee suggests a more detailed discussion of the choice of parameters. While we have provided general results in Sec. 2, we have indeed chosen a particular model system in Sec. 3. Following the comment of the referee, we have extended the discussion of the consequences of the particular choice of an abrupt QPC by a few sentences at the end of the first paragraph of Sec. 3.2.1. However, the quantitative results in Sec. 3 and Sec. 4 are quite general since the electrochemical potential, bias voltage, temperature and dissipated power are expressed in units of the main system parameter (the lowest transverse energy E_1 in the narrow part of the constriction). Only when we discuss the results beyond the qualitative features, we mention estimates of parameter values for the situation of the experiment of Ref. [13] in order to confront our results with the observations.

---

## Round 4 · Author Response

Dear editors,
We have revised and further improved our paper in the light of the comments and suggestions made by the referees. We herewith resubmit our work for publication in SciPost Physics.
Sincerely yours,
Carmen Blaas-Anselmi, Félix Helluin, Rodolfo A. Jalabert, Guillaume Weick, Dietmar Weinmann

---

## Round 4 · List of Changes

List of changes:

  • Following the suggested change 1 of report 1, we have explicitly included a first-order term in the bias voltage in the expansion of the transmission (6), and we discuss its consequences for the expansions (7) and (8) of the dissipated power and the asymmetry below Eq. (8). Moreover, we have moved a sentence about the absence of a first-order correction in bias voltage to the transmission in the case of left-right symmetric systems from a potentially misleading place in the paragraph above Sec. 2.1 to the end of Sec. 2.2.

  • Following the suggested change 2 of report 1, we mentioned the electric asymmetry in the discussion at the end of Sec. 2.2.

  • Following the suggested change 3 of report 1, as well as the suggested change 2 of report 3, we included a discussion of the difference between the result for an abrupt QPC and a smooth QPC model at the end of the first paragraph of Sec. 3.2.1.

  • Following the suggested change 4 of report 1, we alert the reader at the end of the second paragraph of Sec. 3.2.1. about the possible effect of neglected electron-electron interactions at strong bias voltage.

  • We have recently been contacted by colleagues working on the atomic-scale structural fluctuations of an STM plasmonic cavity (new Refs. [41,42]), pointing that the very different behavior obtained under the reverse polarization could be explained by the thermal effects that follow from an asymmetric dissipation. We have then included in the conclusions of the revised manuscript a paragraph commenting on a possible link of our results with those of Refs. [41,42].

  • We have corrected few mistypes and improved some formulations for better readability.

---

## Editorial Decision

published